

# Shift of symbiont communities in *Acropora tenuis* juveniles under heat stress

Makiko Yorifuji[1], Saki Harii[1], Ryota Nakamura[2] and Masayuki Fudo[3]

[1] Sesoko Station, Tropical Biosphere Research Center, University of the Ryukyus, Mobotu, Okinawa, Japan
[2] Fisheries Infrastructure Development Center, Chuo, Tokyo, Japan
[3] Fisheries Agency, Ministry of Agriculture, Forestry and Fisheries, Chiyoda, Tokyo, Japan

## ABSTRACT

Ocean warming is a major threat to coral reefs, leading to an increasing frequency and amplitude of coral bleaching events, where the coral and its algal symbiont associations breakdown. Long-term change and resilience of a symbiont community in coral juveniles is thought to be one of the most important aspects for determining thermal tolerance of the coral holobionts; however, despite its importance, they are not well documented in both under elevated temperature and even under natural condition. Here we investigated changes in symbiont communities in juveniles of the coral *Acropora tenuis* under controlled heat stress conditions (30 °C, 31/32 °C) and natural variations in seawater temperatures (19–30 °C) for up to four months. Compared with the ambient temperature conditions, coral survival rates were higher when exposed to 30 °C, but survival rates decreased when exposed to 31/32 °C. *Symbiodinium* types A3, C1, and D1-4 were detected in the juveniles under all thermal conditions; however, in higher water temperatures (31/32 °C), both the prevalence of D1-4 *Symbiodinium* and the number of juveniles harboring only this type of symbiont increased after two to four months later. In contrast, colonies at lower temperatures (30 °C and ambient) harbored multiple clades of symbionts over the same experimental period. These results highlight the flexibility of the coral–*Symbiodinium* symbiosis for juvenile *A. tenuis* under variable thermal conditions. In particular, the benefit of the preferential association with type D1-4 can be considered as a response when under heat-stress conditions, and that could help corals to cope with ocean warming.

# INTRODUCTION

Coral reefs are currently being threatened by ocean warming and local stressors such as deterioration of water quality and overfishing (*Hoegh-Guldberg, 1999*; *Hoegh-Guldberg et al., 2007*). Increasing seawater temperatures cause coral bleaching (breakdown of the coral and its algal symbiont association), and sometimes mortality (*Hoegh-Guldberg & Smith, 1989*; *Weis et al., 2008*). Massive bleaching events over recent decades have caused damage to many coral reefs (*Hoegh-Guldberg, 1999*; *Hughes et al., 2017*; *Kayanne et al., 2002*; *Loya et al., 2001*) and shifts in the dominant coral taxa have occurred on many coral reefs (*Harii et al., 2014*; *Van Woesik et al., 2011*).

Corresponding authors
Makiko Yorifuji,
makikoyorifuji@gmail.com
Saki Harii, sharii@lab.u-ryukyu.ac.jp

Coral endosymbionts are dinoflagellates of the genus *Symbiodinium*, commonly known as zooxanthellae (*Muscatine & Porter, 1977*). Recent molecular phylogenetic analyses have revealed that the genus *Symbiodinium* contains nine highly divergent genetic lineages, clades A–I (*Pochon & Gates, 2010*). Subsequently, these groups have been further subdivided into numerous subclades or types (*Coffroth & Santos, 2005*).

Facing environmental stressors (e.g., heat, solar irradiance), some of the clades and/or types of *Symbiodinium* exhibit different physiological responses, which contribute to the degree of stress tolerance and/or resilience of the coral holobiont (i.e., coral host + *Symbiodinium*) (*DeSalvo et al., 2010*; *Kinzie III et al., 2001*; *Rowan, 2004*; *Sampayo et al., 2008*; *Suggett et al., 2008*; *Tchernov et al., 2004*). In some scleractinian corals, the relative abundances of certain symbiont genotypes have shifted after bleaching events or high thermal conditions (*Baker et al., 2004*; *Glynn et al., 2001*; *Jones et al., 2008*; *Rowan, 2004*; *Toller, Rowan & Knowlton, 2001*), although this is not always the case (*Stat et al., 2009*). For example, in 1997 in Panama, corals in the genus *Pocillopora*, harboring clade C *Symbiodinium*, were bleached; subsequently, the percentage of corals that harbored clade D increased from 43% to 63% (*Baker et al., 2004*). In 2006 on Keppel Island, part of the Great Barrier Reef, *Acropora millepora* corals, which initially harbored type C2 *Symbiodinium*, changed their symbionts to either type D or C1 after a bleaching event (*Jones et al., 2008*). Another report indicates that *A. millepora* corals changed their symbionts from type C2 to type D after transplantation to a location with higher seawater temperatures (*Berkelmans & Van Oppen, 2006*). In Florida, *Montastraea cavernosa* corals, which normally host type C3 *Symbiodinium*, developed a new association with type D1a *Symbiodinium* after a 10-day treatment at 32 °C (*Silverstein, Cunning & Baker, 2015*). Overall, many corals harboring clade D *Symbiodinium* showed higher thermal tolerance and resistance to bleaching events than those harboring clade C. Although, one report states that *Acropora tenuis* juveniles, artificially infected with type C1 *Symbiodinium*, showed a higher heat tolerance than those infected with type D symbionts (*Abrego et al., 2008*).

Many of the broadcast-spawning corals (about 80%) acquire their symbionts from the surrounding environment (horizontal transmission) during the early stages of their life history, while the other corals acquire symbiotic algae from their parents (vertical transmission) (*Baird, Guest & Willis, 2009*). In the case of horizontal transmitters, juvenile corals can acquire various types of *Symbiodinium*; in contrast, adult colonies are restricted to more specific types of symbionts (*Little, Van Oppen & Willis, 2004*; *Rodriguez-Lanetty, Krupp & Weis, 2004*; *Rodriguez-Lanetty et al., 2006*). For example, in the Great Barrier Reef, *Acropora longicyathus* juveniles harbor clades A, C, or D *Symbiodinium* or a combination thereof, whereas the adults harbor clade A and/or C (*Gómez-Cabrera et al., 2007*). In addition, juveniles of *A. tenuis* and *A. millepora* from the Great Barrier Reef harbor type C1 and/or type D *Symbiodinium*, while the adults typically harbor only one of these types, but not both, as was shown using the nuclear ribosomal DNA internal transcribed spacer 1 (nrITS1) types (*Abrego, Van Oppen & Willis, 2009*; *Little, Van Oppen & Willis, 2004*), although a more recent study has shown that *A. millepora* corals in the Great Barrier Reef harbor both C1 and D types (*Bay et al., 2016*). On Ishigaki Island, Okinawa, types A1, A3, D1, D4, and several types of clade C were detected in several acroporid

juveniles, while the adults, which mainly harbored type C2 (both nrITS1 and ITS2 types) (*Yamashita et al., 2013*; *Yamashita et al., 2014*). Although changes in symbiont type have been well documented for adult corals exposed to high seawater temperature, there is still a gap in our understanding of the responses of coral-*Symbiodinium* associations at the juvenile stage under such stressful conditions. The types of *Symbiodinium* acquired by juvenile corals may change when the corals experience abnormally high seawater temperatures, especially during the summer. Related to this point, *Abrego, Willis & Van Oppen (2012)* examined symbiont types in *A. tenuis* and *A. millepora* juveniles during the onset of symbiosis under high temperature conditions. They found that the juveniles mainly acquired type D *Symbiodinium* at 30 °C and/or 31 °C, whereas only type C1, or an equal number of types C1 and D, were found at 28 °C. Their study showed potential symbiont selection by coral juveniles under different temperature conditions; however, since this experiment was performed over a short period (one month), it is unclear whether the symbiont types acquired by coral juveniles will be maintained in later life stages.

In this study, we investigated changes in the associated *Symbiodinium* communities of coral juveniles under high seawater temperatures for a long-term period of two or four months. In addition, *Symbiodinium* communities in surviving individuals were monitored for up to 1.5 years. We examined *A. tenuis*, which is commonly distributed in the Pacific region (*Veron, 2000*), and is a heat-stress-sensitive species compared with other coral species (*Carpenter et al., 2008*). *A. tenuis* acquires its symbionts through horizontal transmission, and juveniles of the species can obtain multiple types of *Symbiodinium* (*Abrego, Van Oppen & Willis, 2009*; *Little, Van Oppen & Willis, 2004*; *Yamashita et al., 2014*; *Yuyama, Harii & Hidaka, 2012*; *Yuyama et al., 2005*).

## MATERIALS AND METHODS

### Preparation of coral juveniles

Parental *A. tenuis* colonies were collected in 2010 from Aka Island, Okinawa, Japan (26.19°N, 127.29°E), and were maintained in an outdoor tank with coarse sand filtered (sand diameter 1.7–2.8 mm) running seawater on the island. The same set of four parental coral colonies spawned on June 10, 2012 and June 1, 2013; their gametes were mixed to allow fertilization to occur and resulting embryos were cultured until they developed into planula larvae. Planulae were subsequently transferred to new tanks with preconditioned ceramic tiles for settlement. Tiles were 10 cm × 10 cm in size, and were soaked in the sea off the island for two months prior to coral spawning, allowing for the development of natural biofilm layers. After larval settlement, the tiles were kept in an outdoor tank with coarse filtered running seawater to promote *Symbiodinium* acquisition from natural seawater until the experiment commenced.

### Temperature treatments and collection of coral juveniles

The coral juveniles were transferred to a set of experimental tanks on Aka Island one month after settlement in 2012 and two months after settlement in 2013. Three seawater temperature conditions were used: (i) ambient seawater ranged 22.66–30.82 °C in 2012 and 27.01–30.93 °C in 2013 (Fig. S1); (ii) moderate heat-stress temperatures of 30 °C;

and (iii) high heat-stress temperatures of 31 °C in 2012 or 32 °C in 2013. Two and three independent tanks for each temperature condition were prepared in 2012 and 2013, respectively (Fig. S2). The heat temperature conditions in the experimental tanks were regulated and controlled by heaters (TH2-1; Nittokizai Co. Ltd., Saitama, Japan) and thermostats (RL-200N; Marugo, Japan). The water temperatures were recorded hourly by thermo loggers during experimental periods (HOBO Water Temp Pro v2 U22-001; Onset Computer Corporation, Bourne, MA, USA); one logger was set in one of each temperature treatment tanks. The average ± SD of recorded temperatures in the heated experimental tanks were 29.78 ± 0.42 °C (for 30 °C) and 30.83 ± 0.89 °C (for 31 °C) in 2012, and between 30.06 ± 0.17 °C (for 30 °C) and 32.06 ± 0.34 °C (for 32 °C) in 2013. The tanks were placed under natural sunlight, which was reduced with a transparent plastic roof and a black mesh. The daily maximum light intensity (measured as photosynthetic active radiation, PAR) in the tanks was measured by a light intensity logger (ALW-CMP, JFE Advantech, Nishinomiya, Japan) for approximately one week each month during the experiment. The average daily maximum light intensity during the experiments was 196.8 $\mu$mol photon/m$^2$/s (range 41.1–436.3 $\mu$mol photon/m$^2$/s). In each tank, 250–400 colonies of coral juveniles were reared on six to ten tiles. All experimental tanks were 100 L in volume and were supplied with 10 L coarse filtered running seawater every hour. Because coral juveniles in 2012 survived exceedingly well for up to four months, even at 31 °C with two replicate tanks for each condition, experimental conditions were changed slightly between the two years. In 2013, 32 °C of high heat stress and three replicate tanks were set in order to provide clearer results.

Coral juveniles were observed and collected after two weeks, and then monthly for four months after the heat experiment began in 2012. In 2013, coral juveniles were collected and observed after one, two, three, and five months. In 2012, juveniles that survived more than four months were continuously reared in the temperature-maintained tanks and collected 1.5 years later. On each observation date, the numbers of live individuals (colonies) were visually counted for survivorship, and then 10–30 total individuals including several backup specimens were collected from each tank (randomly from several tiles) to measure growth and to identify the associated symbiont communities. Growth of the juveniles was measured by the size of their skeletons. The major and minor axes of their undersides were measured and the geometric mean diameter (GMD: square root of the product of the two numbers) was calculated. Juveniles were then fixed in 99.5% ethanol for genetic analyses. The tips of the branches (approximately 1–2 cm) of the four adult parent corals that were used to obtain gametes were also collected and fixed for genetic analyses in November 2012.

### Genetic analyses of *Symbiodinium* communities

Total DNA from both juvenile and adult coral samples was extracted with guanidine solution using the method described by *Sinniger, Reimer & Pawlowski (2010)*. The clades/types of *Symbiodinium* spp. were identified by comparing differences in their nrITS2 sequences. Denaturing gradient gel electrophoresis (DGGE) was used to separate the PCR products. Since coral juveniles are small and the density of symbionts in
some colonies was expected to be low, nested PCR was conducted to obtain target DNA fragments as follows: the nrITS region was amplified in the first reaction using the primer pair ZITSUPM13/ZITSDNM13 (*Santos, Taylor & Coffroth, 2001*) and the amplification conditions described by the designers. In the second reaction, the nrITS2 region was amplified using the first PCR product with primer pair ITSint-for2/ITS2CLAMP (*LaJeunesse & Trench, 2000*). The amplification conditions were described by *LaJeunesse et al. (2003)*. The amplified nrITS2 PCR products were separated by DGGE as described by *LaJeunesse & Trench (2000)* with modifications following *Yorifuji et al. (2015)*, using the Bio-Rad DCode System (Bio-Rad Laboratories, Inc., Hercules, CA, USA).

To identify the clades and types of associated *Symbiodinium* spp., unique DGGE bands were selected from their profiles (derived from 29 colonies in 2012 and 44 colonies in 2013) and sequenced. DNA fragments in the target DGGE bands were amplified using the primers ITSint-for2 and ITS2 rev (reverse primer without the GC-rich clamp) with the following thermal cycling profile: initial denaturation at 94 °C for 5 min; 35 cycles of denaturation at 94 °C for 45 s, annealing at 52 °C for 45 s, and extension at 72 °C for 60 s; and a final extension at 72 °C for 10 min. These PCR products were purified and sequenced by the Sanger (dye-terminator) method performed by Macrogen Japan (http://www.macrogen-japan.co.jp/) using the primer pair ITSint-for2/ITS2 rev. The obtained sequence data were edited and assembled using ATSQ, version 6.0.1 (Genetyx Co., Tokyo, Japan), and identified through BLAST searches of the International Nucleotide Sequence Database (INSD) and the GeoSymbio database (*Franklin et al., 2012*), which provides alignments of *Symbiodinium* nrITS2 sequences that were published between 1982 and 2012.

## Statistical analyses

Growth of the corals was estimated from the mean values of their skeleton sizes in each experimental tank and then compared across temperature conditions using the Wald test under a generalized linear model (GLM). Survival curves of the coral juveniles were estimated using survival curve models with the Kaplan–Meier method and compared across temperature conditions using the two-sided log-rank test.

The coral-*Symbiodinium* partnership patterns, as a function of different temperature treatments, were evaluated through the DGGE profiles obtained. The prevalence of the *Symbiodinium* types and the composition of the symbionts were used as indicators of the symbiont communities, since some corals harbored a single type of *Symbiodinium*, while others harbored multiple types. The percentage of coral juveniles harbored each *Symbiodinium* type was calculated based on the number of host colonies analyzed under each temperature condition (referred to as "prevalence" of symbiont type). The symbiont composition for each host colony was represented by the DGGE profile (band pattern) obtained. The number of individual hosts at each temperature that harboring the same symbionts was counted based on the DGGE profiles, and the proportions of juveniles with similar symbiont compositions at each temperature were calculated. Comparisons of symbiont prevalence and symbiont composition at the various temperatures were made separately for the colonies collected at the beginning and end of the experiment for each

year, using the Fisher's exact test. Subsequently, a cluster analysis was performed to evaluate the similarities of symbiont compositions under the three temperature treatments over the experimental time course. Sets of Euclidian distances between treatments and sampling times were calculated from the juvenile proportions of the symbiont composition data. A hierarchical cluster analysis was conducted using the unweighted pair-group method and the arithmetic mean based on Euclidian distances. All statistical analyses above were performed using the statistical package R, version 3.0.2 (*R Core Team, 2013*).

## RESULTS

### Survivorship and growth of coral juveniles

The *A. tenuis* juveniles in all the treatments survived for up to four months after the experiment began in 2012 (Fig. 1A). Survival rates were 24.4% (ambient) and 14.0% (31 °C) at four months. Corals in the 30 °C tanks survived up to 1.5 years, with a 37.0% survivorship at four months. In 2013, coral juveniles in the 32 °C and ambient tanks survived only two months (Fig. 1B). Survival rates were 15.2% (ambient), 51.3% (30 °C), and 3.4% (32 °C) at two months. Survivorship in both heated treatments was significantly different from ambient survivorship for both years ($P < 0.001$ by the two-sided log-rank test).

The GMDs of the coral juveniles in 2012 ranged from $1.3 \pm 0.28$ mm (ambient) to $1.5 \pm 0.35$ mm (31 °C) on the first day of temperature treatments; four months later, the GMDs ranged from $3.4 \pm 1.06$ mm (31 °C) to $3.9 \pm 0.87$ mm (ambient) (Fig. 2A). The growth of the corals under heated conditions was not significantly different from that of the ambient treatment group ($P = 0.09$ at 30 °C, $P = 0.25$ at 31 °C, Wald test under GLM). Sizes in 2013 ranged from $1.5 \pm 0.40$ mm (32 °C) to $1.8 \pm 0.39$ mm (ambient) on the first day of the experiment; two months later, sizes ranged from $1.8 \pm 0.50$ mm (32 °C) to $2.2 \pm 0.43$ mm (ambient) (Fig. 2B). The growth rates of the corals under the heated conditions were not significantly different from those of the ambient group ($P = 0.32$ at 30 °C, $P = 0.86$ at 32 °C, Wald test under GLM).

### Symbiont types in coral juveniles and adults

In coral juveniles at all temperatures, four major bands were detected in the DGGE gels (Fig. S3), and these four bands corresponded to four specific sequences of *Symbiodinium*—A3, C1, D1, and D1a—according to the nrITS2 sequences (identical to INSD sequences AF333507, AF333515, AF334660 (*LaJeunesse, 2001*), and AF499802 (*LaJeunesse, 2002*), respectively). In addition, a novel type of clade F *Symbiodinium* was detected from one colony that was reared in an ambient tank one month after the experiment began in 2012. This sequence was deposited in the DNA Data Bank of Japan (DDBJ) under accession number LC015663. All four parental corals harbored only C3 *Symbiodinium* (AF499789 (*LaJeunesse, 2002*)). In coral juveniles, sequences of D1 and D1a *Symbiodinium* were obtained together from each individual; thus, these sequences were considered to be derived from type D1-4, later described as *Symbiodinium trenchii* (*LaJeunesse et al., 2010*; *LaJeunesse et al., 2014*). For further analyses, clade F *Symbiodinium* was removed as it was found only once throughout the experiments.

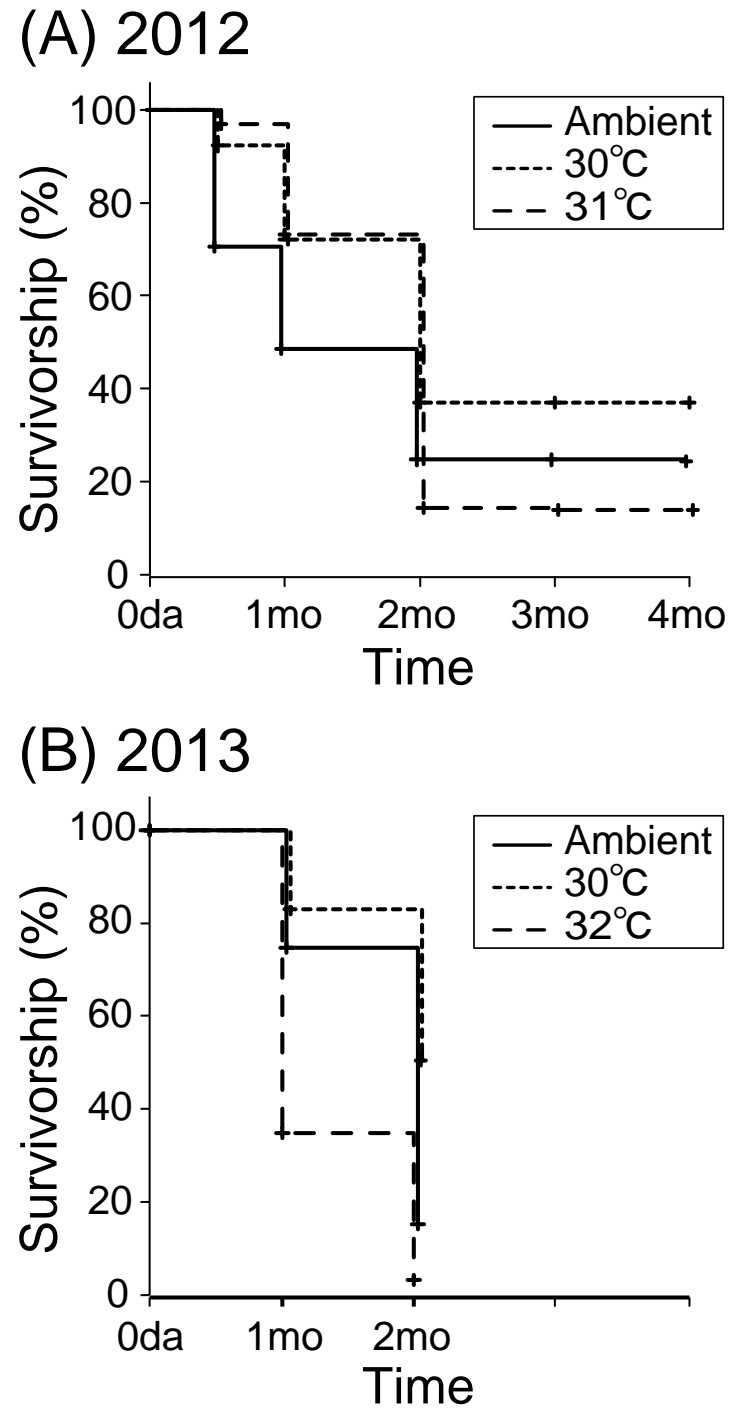

**Figure 1  Survival rates for juveniles of *Acropora tenuis*.** Numbers of live colonies were counted, and survival curves were obtained using the Kaplan–Meier method for years 2012 (A) and 2013 (B).

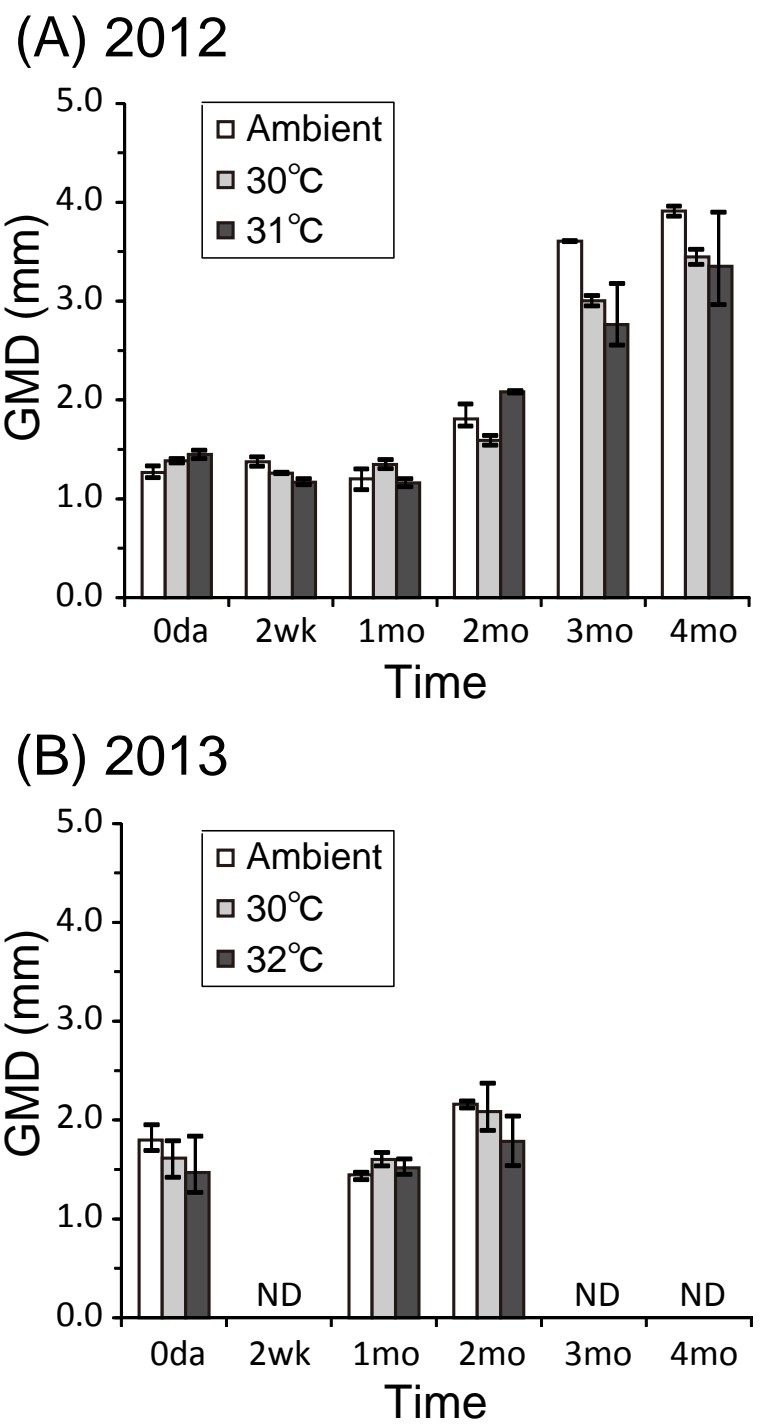

**Figure 2 Growth of *Acropora tenuis* juveniles.** Growth of *Acropora tenuis* juveniles. Mean values of the GMDs (see Materials and Methods) within temperature treatment groups are shown for years 2012 (A) and 2013 (B). Bars indicate the maximum and minimum mean values observed among replicates (two in 2012 and three in 2013). ND, no data.

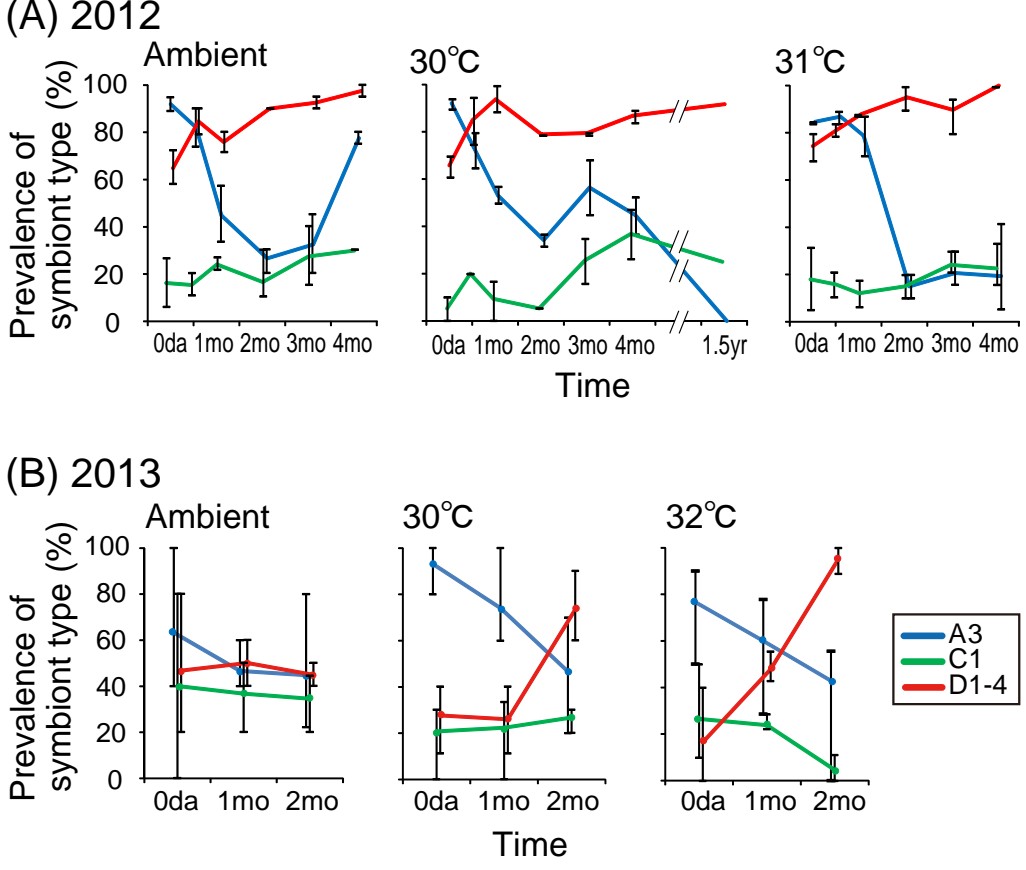

**Figure 3** **Prevalence of *Symbiodinium* types in *Acropora tenuis* juveniles.** Prevalence of *Symbiodinium* types in *Acropora tenuis* juveniles. The percentage of juveniles that harboring each *Symbiodinium* type was calculated for each temperature treatment group in 2012 (A) and in 2013 (B). Bars indicate the maximum and minimum values of the percentage observed in each replicate (two in 2012 and three in 2013). Numbers of individuals analyzed are shown in Table S1.

## Changes in *Symbiodinium* prevalence among coral juveniles

The prevalence of *Symbiodinium* types associated with coral juveniles displayed variable patterns according to the environmental conditions. Generally, the prevalence of type A3 *Symbiodinium* was high at the beginning of the experiment and then decreased; the prevalence of type C1 remained low and increased only at 30 °C; and the prevalence of type D1-4 increased in both years (Table S1, Fig. 3).

In 2012, the prevalence of type A3 in juveniles was more than 80.0% at all temperature conditions initially, but decreased to 15.0%–34.2% after two months. It increased again to 77.5% after four months at ambient and 30 °C conditions, but remained around 20% at 31 °C (Fig. 3A). The prevalence of type C1 was low (less than 20%) initially at all temperatures, and then gradually increased up to 36.8% during the four-month experiment. The range of increase was about 30% at 30 °C, but less than 15% under ambient temperatures and at 31 °C. The prevalence of type D1-4 remained high and

gradually increased during the four-month experimental period at all temperatures. The prevalence of the three *Symbiodinium* types A3, C1 and D1 did not significantly differ at the beginning of the experiment among the different temperature conditions (A3: $P = 0.621$, C1: $P = 0.134$, D1-4: $P = 0.610$, Fisher's exact test); however, the prevalence of the three types was significantly different after four months (A3: $P < 0.001$, C1: $P < 0.001$; D1-4: $P = 0.028$, Fisher's exact test). For colonies exposed to temperature conditions of 30 °C, the prevalence of types A3, C1, and D1-4 were 0.0%, 25.0%, and 91.7%, respectively, 1.5 years after the experiment began.

In 2013, the prevalence of type A3 was initially high (63.3%–96.4%) in juvenile corals at all temperatures, and then decreased to 41.4%–46.7% (Fig. 3B). The prevalence of type C1 remained consistently around or lower than 40% under all temperature conditions. The prevalence of type D1-4 increased in the two heated treatments; the amounts of the increases at 32 °C varied widely, ranging from 17.2% to 95.2% over two months. The prevalence of *Symbiodinium* types did not significantly differ among temperature conditions at the beginning or after two months (A3: $P = 0.960$ (2 months); C1: $P = 0.300$ (0 day); $P = 0.411$ (2 months); D1-4: $P = 0.051$), but the prevalence of type A3 at the beginning and type D1-4 after two months differed significantly ($P = 0.006$ and $P < 0.001$, respectively, Fisher's exact test).

### Changes in *Symbiodinium* type compositions in each coral juvenile

One or multiple *Symbiodinium* clades/types were found in each individual colony, and these *Symbiodinium* compositions also changed during the experimental course (Table S2, Fig. 4 and Fig. S4).

In 2012, 50%–60% of coral juveniles at all temperatures harbored both types A3 and D1-4 simultaneously after two weeks to one month (Fig. 4A). After two months, the number of colonies that harbored only type D1-4 *Symbiodinium* increased to approximately 60%. After three to four months, under ambient and 30 °C conditions, colonies harboring only type D1-4 decreased to fewer than 20%, while those harboring multiple clades/types of symbionts increased. In contrast, the number of corals harboring only type D1-4 remained high (55%–65%) at 31 °C. Additionally, more than 92% of the colonies at 30 °C harbored only type D1-4 after 1.5 years (Fig. 4A). Under all temperature conditions, symbiont compositions after four months and 1.5 years were significantly different from those at the beginning of the experiment (Table 1). The compositions in the heated treatments after four months were significantly different from those under ambient conditions.

In 2013, 50%–70% of the coral juveniles at all temperatures harbored type A3 or C1 *Symbiodinium* at the beginning of the experiment (Fig. 4B). After two months at 32 °C, colonies harboring only type D1-4 *Symbiodinium* increased to 52% conditions (Table S2, Fig. 4B). In heated treatments, the symbiont compositions after two months were significantly different from those at the beginning of the experiment; no significant changes were observed under ambient conditions (Table 1). Among the temperature treatments, symbiont composition was significantly different only between the 32 °C and ambient conditions. A greater proportion of surviving coral juveniles harbored only D1-4 *Symbiodinium* under 31/32 °C conditions compared with the other treatments. This

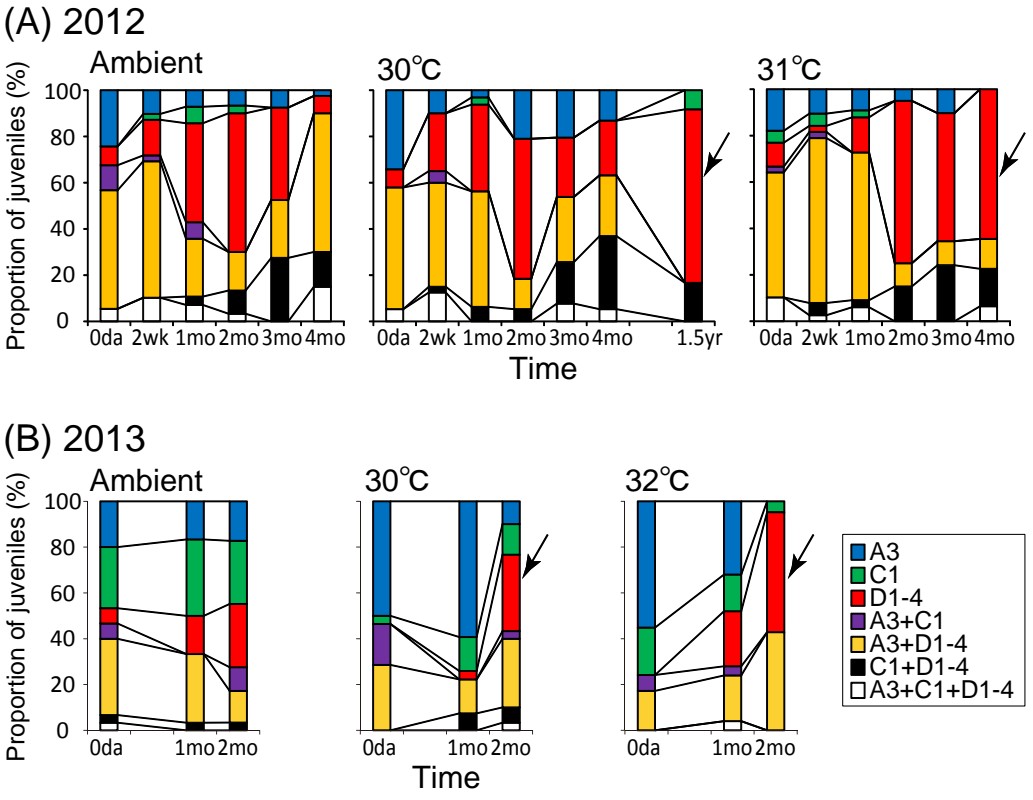

**Figure 4** **Symbiont compositions and their occurrences.** *Symbiodinium* type compositions within individual colonies of *Acropora tenuis* juveniles and proportion of juveniles with the same set of symbionts in each temperature treatment group in 2012 (A) and in 2013 (B). Numbers of individuals analyzed are shown in Table S2. Arrows indicate the increase in coral juveniles bearing only D1-4 bearing coral juveniles under heated conditions.

difference was especially visible when considereing the survivorship of coral juveniles (Fig. S4).

When we statistically compared the similarities of the symbiont compositions among temperature conditions and time (based on data shown in Table S2), three major clusters formed from the 2012 data (Fig. 5A). One cluster primarily contained data obtained during the early stages of the experiment (cluster I), whereas the other two clusters contained data from the later stages of the experiment. One of these two clusters contained data from the ambient and 30 °C conditions during both the earlier and later stages (cluster II). The other cluster primarily contained data from the 31 ° C treatment during the later stages (cluster III; Fig. 5A). In 2013, three major clusters also formed (Fig. 5B). The data obtained from the 32 °C treatment after two months (cluster II) did not cluster with the other data.

## DISCUSSION

This study showed, for the first time, differential survivorships and changes in the associated *Symbiodinium* community composition in juveniles of the scleractinian coral, *A. tenuis*,

**Table 1 Comparisons of *Symbiodinium* type compositions in *Acropora tenuis* juveniles by Fisher's exact test.** Number of star indicates significance of differences.

| Year | Date | Treatment | P-value |
|------|------|-----------|---------|
| 2012 | 0 da vs. 4 mo | Ambient | 0.0006** |
|      |               | 30 °C | <0.001*** |
|      |               | 31 °C | <0.001*** |
|      | 0 da vs. 1.5 yr | 30 °C | <0.001*** |
|      | 0 da | Ambient vs. 30 °C | 0.244 |
|      |      | Ambient vs. 31 °C | 0.435 |
|      | 4 mo | Ambient vs. 30 °C | 0.003* |
|      |      | Ambient vs. 31 °C | <0.001*** |
| 2013 | 0 da vs. 2 mo | Ambient | 0.238 |
|      |               | 30 °C | <0.001*** |
|      |               | 32 °C | <0.001*** |
|      | 0 da | Ambient vs. 30 °C | 0.019 |
|      |      | Ambient vs. 32 °C | 0.060 |
|      | 2 mo | Ambient vs. 30 °C | 0.410 |
|      |      | Ambient vs. 32 °C | 0.003* |

when exposed to high water temperatures over a long period of time (up to four months). Low survivorship of *A. tenuis* juveniles was observed at the highest temperature treatment (31/32 °C). It is contrasting that coral juveniles survived up to four months even at 31 °C in 2012, but in 2013 those juveniles reared at 32 °C survived for only two months in 2013. This may indicate that the lethal temperature threshold for *A. tenuis* juveniles in Okinawa is at 32 °C. In contrast to the highest temperature treatment, high coral survivorship was observed at 30 °C. Coral bleaching occurs when water temperatures higher than the average summer maximum sea surface temperature persist for an extended period (thermal stress can be estimated using the Degree Heating Week metric) (*Glynn, 1996*; *Logan et al., 2012*). In this study, the average temperatures in the 30 °C tanks were 0.50–1.28 °C above the average ambient temperature during the warmest month (August) in this region. Theoretically, a temperature increase of this magnitude persisting for more than two months could have triggered coral bleaching (*Kayanne, 2017*), however we did not see signs of bleaching. Survivorship in the ambient treatment was lower than at 30 °C. This is perhaps because the temperature was not stable under ambient conditions. Temperatures fluctuated daily, including sudden changes caused by typhoons, and then a decrease decreased in winter. In contrast to survivorship, no significant differences in coral growth were observed between the temperature treatments, suggesting that water temperature does not affect coral growth during the early life stages.

*A. tenuis* juveniles in this study harbored a greater variety of *Symbiodinium* types compared with adults; juveniles harbored types A3, C1, D1-4, and F, whereas adults harbored only type C3. This result is consistent with those of previous studies on *Acropora* spp., showing more flexibility during early ontogeny (*Abrego, Van Oppen & Willis, 2009*; *Abrego, Willis & Van Oppen, 2012*; *Gómez-Cabrera et al., 2007*; *Little, Van Oppen & Willis, 2004*; *Yamashita et al., 2013*; *Yamashita et al., 2014*). Even within the species studied here,

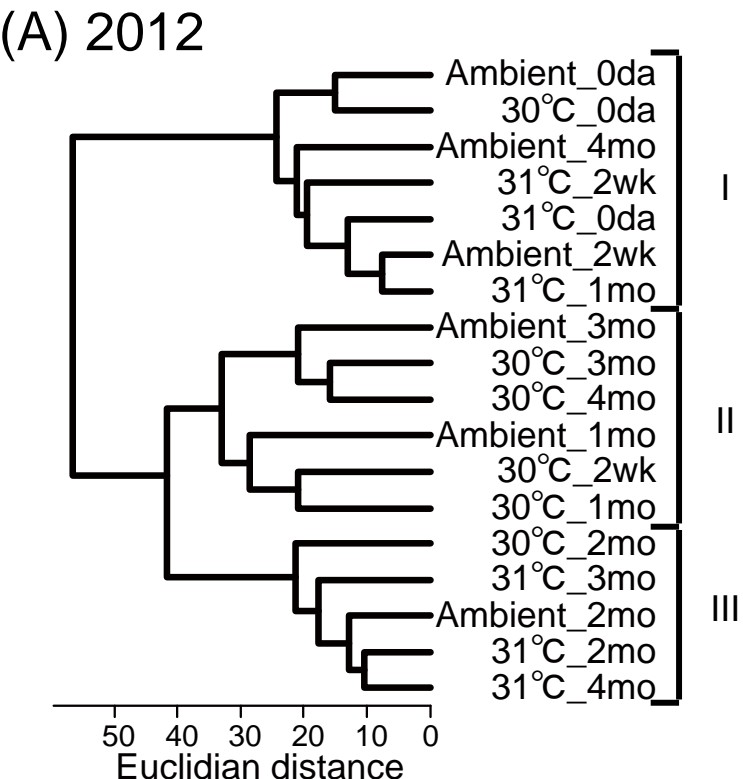

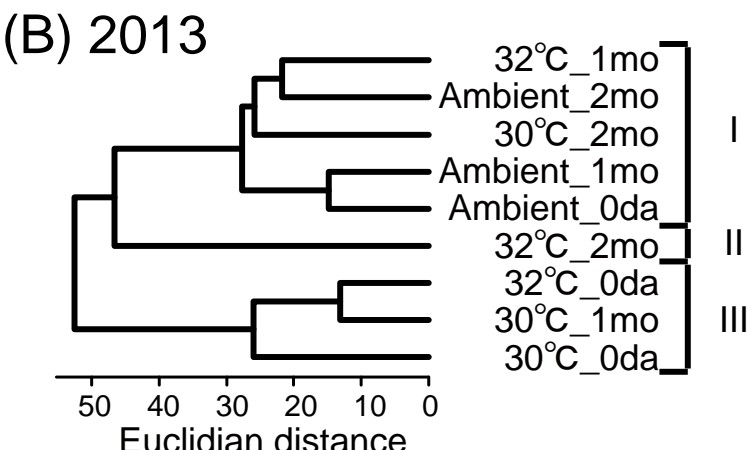

**Figure 5** **Similarities between compositions of detected *Symbiodinium* types in coral juveniles of *Acropora tenuis* under three temperature treatments.** Similarities were evaluated for years 2012 (A) and 2013 (B) by cluster analyses based on the proportions of juveniles with similar symbiont compositions (data shown in Table S2). I to III indicate clusters detected.

*A. tenuis*, distinct symbiont communities were observed in different locations. In the Great Barrier Reef, juveniles of *A. tenuis* initially acquired *Symbiodinium* C1, D, or a combination of both (*Abrego, Van Oppen & Willis, 2009*; *Abrego, Willis & Van Oppen, 2012*; *Little, Van Oppen & Willis, 2004*), whereas on Ishigaki Island, southern Japan, these corals hosted clades A, C, D, or a combination of these types, one to six months after settlement (*Yamashita et al., 2013*). Moreover, in our study, we found for the first time the presence of clade F *Symbiodinium* in *A. tenuis*. These differences in acquired *Symbiodinium* types are believed to be caused by locally-available *Symbiodinium* communities in sediments and seawater, where free-living algae play roles in the onset of symbiosis in horizontal transmitters for the acquisition of symbionts (*Adams, Cumbo & Takabayashi, 2009*; *Cumbo, Baird & Van Oppen, 2012*; *Nitschke, Davy & Ward, 2015*; *Yamashita et al., 2013*). In this study, *Symbiodinium* communities in coral juveniles differed between the two years, and this may reflect several factors that differ from year to year. The genetics of juveniles may differ by cohorts and that resulted in differences in symbiont preferences of the juveniles. Water temperatures in ambient tanks were also different among the two years (see Fig. S1), and temperatures higher than 30.5 °C were observed more frequently in the ambient tanks during 2013. In addition, the number of typhoons that approached Okinawa in 2012 was double the number of those in 2013 (eight and four, respectively; Japan Meteorological Agency). Excessive disturbances may have led to differences in factors such as increased water temperature fluctuations and may also in plankton flora in the water column. *Symbiodinium* in the water column may also differ. *Yamashita et al. (2013)* reported that free-living *Symbiodinium* communities in the water column change over time, and that their compositions were not the same even within the same season.

In the present study, the percentage corals harboring only D1-4 *Symbiodinium* at the highest temperature treatments (31/32 °C) increased over time (Fig. 4). Even when the proportions of the symbiont composition were calculated based on coral survivorship, the number of colonies hosting only type D1-4 seemed stable at higher temperatures, particularly in 2012 (Fig. S4). Although the trend in increase was different in 2013, but the prevalence of D1-4 was strikingly increased in higher temperature (Fig. 3B). An increase in only type D1-4-bearing corals was also observed under ambient conditions two months later in 2012, and these symbiont communities were statistically similar to those reared under higher temperatures in later stages of the experiment (Fig. 5). These results may be because the ambient water temperature was sometimes higher than 30 °C at the time. Clade D *Symbiodinium* are known to provide heat tolerance to adult corals (*Abrego et al., 2008*; *Baker et al., 2004*; *Berkelmans & Van Oppen, 2006*; *Fabricius et al., 2004*; *Glynn et al., 2001*; *LaJeunesse et al., 2008*; *Rowan, 2004*; *Stat & Gates, 2011*; *Toller, Rowan & Knowlton, 2001*). In juvenile corals, clade D-bearing colonies of *A. tenuis* were thermally less robust when compared with those harboring type C1 in short-term experiments (*Abrego et al., 2008*). However, the uptake of clade D *Symbiodinium* increased under heated conditions (30 °C and 31 °C) during the onset of symbiosis in *A. tenuis* and *A. millepora* (*Abrego, Willis & Van Oppen, 2012*). Moreover, juveniles of *A. tenuis* experimentally infected with clade D *Symbiodinium* exhibited a higher survival rate (*Yuyama et al., 2016*) and higher carbon acquisition (*Baker et al., 2013*) than those infected with clade C1 at 30 °C. Therefore, in our

study, the increase in juveniles bearing only type D1-4 at higher temperatures suggests that type D1-4 contributes to the heat tolerance of the juveniles. It is also possible that juveniles hosting a mix of clades, including type D1-4, survived but lost the other clades and/or that they changed their symbiont from other types to D1-4 (symbiont shuffling/switching). In 2012, the prevalence of type D1-4 was stayed high during the experimental period, but those of A3 and C1 decreased at highest temperature may exhibit this possibility. Such possibilities have been reported for adults that have experienced bleaching events (*Jones et al., 2008*; *Silverstein, Cunning & Baker, 2015*), and is termed the "adaptive bleaching hypothesis" (*Buddemeier et al., 2004*). The differential changes in associated *Symbiodinium* community composition relating to temperature conditions, as observed in this study, indicate a potential flexibility in changing symbionts. Further studies are required to test this hypothesis by monitoring the same coral individuals over time; however, this approach is challenging due to the small size of the coral juveniles.

The prevalence of A3 *Symbiodinium* was high at the beginning of the experiments. A3 *Symbiodinium* is also considered a possible critical symbiotic partner during the early life stages of *A. tenuis* in the Great Barrier Reef, as more robust strains of corals harbor higher proportions of *Symbiodinium* types A3 and D1 (*Quigley, Willis & Bay, 2016*). Therefore, this type of symbiont also plays an important role in *A. tenuis* survival in Okinawa. A3 *Symbiodinium* showed a rapid increase under ambient conditions during later periods in the experiment (i.e., winter) in 2012. This finding suggests a cold adaptation of the holobionts. Although decreases in A3 *Symbiodinium* were observed at higher temperatures during later stages of the experiments, the changes during early stages and under ambient conditions differed between the two years. These differences may be caused by variable environmental conditions, such as fluctuating ambient seawater temperatures between 2012 and 2013 under ambient conditions differed among both years (Fig. S1).

Most of the coral juveniles surviving after 1.5 years at 30 °C harbored clade D *Symbiodinium*. This is in contrast to *A. tenuis* adults in the northwestern Pacific that are normally associated with clade C *Symbiodinium* (*LaJeunesse et al., 2004*; this study), although no comparisons can be made between ambient and heated treatments in the present study. During coral growth, the symbiont communities of the coral juveniles become more specialized (*Abrego, Van Oppen & Willis, 2009*; *Gómez-Cabrera et al., 2007*; *Little, Van Oppen & Willis, 2004*; *Yamashita et al., 2013*). *Abrego, Van Oppen & Willis (2009)* reported that the symbiont communities of *A. tenuis* were homologous to those of adults by 3.5 years of age, and significant changes in the adult-hosted symbiont types were detected by 1.5 years of age in comparison with corals one month after settlement. In contrast, none of the juveniles in this study harbored the C3 *Symbiodinium* that was harbored by their parents, even 1.5 years later. This may be because our juvenile corals in tanks had no access to the C3 symbiont including those expelled from adult colonies, but it also may be due to heat treatments altering the coral preference for their symbiont.

Since many corals spawn in late spring to early summer (*Harrison, 2011*; *Hayashibara et al., 1993*), their new recruits experience the highest seawater temperatures in any given year. Recently, the frequency of extreme surface seawater temperature events in summer has increased (*Hughes et al., 2017*; *Kayanne, 2017*). Moreover, ocean temperatures are

gradually increasing and recent simulations indicate that under the worst scenario, sea surface temperatures in the 2090s will be 2.73 ± 0.72 °C higher than those experienced in the 1990s (*Bopp et al., 2013*). Thus, coral recruits will likely face higher summer seawater temperatures in the future, and hosting multiple symbiont types is probably advantageous to juveniles. Indeed, our results show that coral juveniles may be able to adapt to future climate change by changing their symbiont types, resulting in adult *A. tenuis* hosting more thermally tolerant symbionts. Other adult corals, such as pocilloporids, acroporids, and fungiids, did not change their symbiont types under high thermal conditions in the southern Great Barrier Reef in 2002 (*Stat et al., 2009*), however, the 2002 bleaching event was minor in that region. Therefore, the degree of bleaching is likely dependent upon the combined effects throughout the year, the local environmental conditions, and the species ecological/biological traits. This study focused on symbiont community change, but other factors, such as host coral genetics (*Dixon et al., 2015*; *Palumbi et al., 2014*), play roles in the stress tolerance of corals. More studies are needed using different coral genera, different environmental conditions, and physiological/genetic traits to better understand how coral juveniles can acclimatize and/or adapt in the future.

## ACKNOWLEDGEMENTS

We thank Mr. T Nakajima, Mr. A Nakamura, Mr. T Kijima, Ms. S Okitsu, Ms. A Muto, Mr. Y Nakatsuji, and Mr. M Jinza for their kind help in experimental setup, rearing corals, and molecular analyses. We thank Dr. H Rouzé and Dr. F Sinniger for fruitful discussion and comments on the manuscript. Machines for DGGE analyses were kindly provided by Dr. K Inoue from Atmosphere and Ocean Research Institute, Univ. Tokyo and Dr. N Shinzato from Center of Molecular Biosciences, Univ. Ryukyus.

### Funding

This study is a part of the Coral Propagation under Severe Environmental Conditions Project of the Fisheries Agency, Japan. This study was also supported in part by Grant-in-Aid for Scientific Research (C) (No. 24570030) and (A) (No. 16H02490) of the Ministry of Education, Culture, Sports, Science and Technology, Japan and in part of JST Core Research for Evolutional Science and Technology (CREST) Grant Number JPMJCR13A1, Japan to Saki Harii. There was no additional external funding received for this study. The funders had no role in study design, data collection and analysis, decision to publish, or preparation of the manuscript.

### Grant Disclosures

The following grant information was disclosed by the authors:
Fisheries Agency, Japan.
Ministry of Education, Culture, Sports, Science and Technology, Japan: 24570030, 16H02490.
JST Core Research for Evolutional Science and Technology (CREST): JPMJCR13A1.
## Competing Interests

The authors declare there are no competing interests.

## Author Contributions

- Makiko Yorifuji conceived and designed the experiments, performed the experiments, analyzed the data, contributed reagents/materials/analysis tools, wrote the paper, prepared figures and/or tables, reviewed drafts of the paper.
- Saki Harii conceived and designed the experiments, contributed reagents/materials/-analysis tools, wrote the paper, prepared figures and/or tables, reviewed drafts of the paper.
- Ryota Nakamura performed the experiments, contributed reagents/materials/analysis tools, reviewed drafts of the paper.
- Masayuki Fudo contributed reagents/materials/analysis tools, reviewed drafts of the paper.

## DNA Deposition

The following information was supplied regarding the deposition of DNA sequences:
Sequence of a novel type of clade F Symbiodinium detected in this study was deposited in the DNA Data Bank of Japan (DDBJ) under accession number LC015663.

## Supplemental Information

Supplemental information for this article can be found online at http://dx.doi.org/10.7717/peerj.4055#supplemental-information.

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
