# Peer review of "Shift of symbiont communities in Acropora tenuis juveniles under heat stress"

_PeerJ, doi:10.7717/peerj.4055_

## Round 0.1 · original submission · Major Revisions

While the experimental approach is sound (though in need of revisions), it appears that the authors have overstated the role of Symbiodinium in A. tenuis fitness during heat stress. In fact, the data can be interpreted to really only indicate higher prevalence of the D1-4 type in juveniles across both sample points, not necessarily a direct impact of this symbiont on fitness. The relatively low sampling to support the claims reached in the paper should also be considered. All three reviewers raise valid points regarding the interpretation of the results (particularly reviewers 2 and 3) and these should be carefully considered and appropriately addressed in your revision. All three authors also made excellent suggestions to improve the overall quality of the manuscript, including the writing and presentation. I would like to see a full response to these suggestions and their incorporation into the revised manuscript. Overall, I firmly believe this work will greatly benefit from these revisions, and will likely be publishable in PeerJ after completion.

Reviewer 1 ·

Basic reporting

1. The ms is clearly written. Literatures and background are well provided. Figures are acceptable.

Experimental design

Research question well defined and it is stated how this study fills an identified knowledge gap. Investigation performed to a high technical and ethical standard. Replicate should be described more clearly (see the general comments for the authors).

Validity of the findings

Data is robust, statistically sound, but probably small mistakes in description of the results (see the general comments for the authors).

Additional comments

1. The authors wrote in the Introduction that they studied changes in Symbiodinium communities associated with A. tenuis juveniles for the long-period of 1.5 years (line 97-98). But this is misleading, since actually most experiments finished in 2 or 4 months and only small numbers of juveniles survived up to 1.5 years.
2. The authors described that the range of seawater temperature in tanks (the first paragraph of the Temperature treatment section). Do the values mean daily minimum and mean daily maximum temperatures ± SD for 2 tanks in 2012 and for 3 tanks in 2013? Please explain this more clearly.
3. They compared prevalence of Symbiodinium types among temperature treatments. Is P=0.028 for D1-4 after 4 months not significant (line 251)? What is the level of significance used in this study? Is P=0.006 for A3 (0 day) not significant (line 259-262)? Please check.
4. In 2012 experiment, there was a significant difference in the prevalence of the three Symbiodinium types among temperatures. Still the authors describe the changes of each Symbiodinium type at all temperature conditions together (line 243-248).
5. In 2013 experiments, they found no significant effects of temperature conditions on the prevalence of Symbiodinium types. But they described changes in the prevalence of each type separately at each temperature (line 255-259). It is confusing. The meaning of the latter half of the sentence (line 257-259) is not clear.
6. When they state that type C1 decreased under ambient and 32oC, but increased at 30oC (line 255-257), were there significant differences in the prevalence among different sampling time point?
7. They used two tanks for each temperature condition in 2012 and three tanks in 2013 experiment. Do they use individual juvenile as statistical unit regardless of tanks? They described in the method section that 10-20 juveniles were sampled from each tank at each sampling point. But, if they show the number of replicates in the legend of Figs. 2, 3, and 4, it would be clearer.
8. In the discussion, they state that the observation by Abrego et al. (2009) explains why none of the juveniles in this study harbored C3 Symbiodinium. But it is not clear why? Please make it clear. (line 378-380)
9. They state that similar changes within the symbiont community were observed at higher temperature during later stages (line 366-369). But increase of A3 Symbiodinium as seen under the ambient conditions was not observed in high temperature conditions during later stages (Fig. 3). Please clarify this sentence.
10. The authors stated that colonies harboring only D1-4 increased under ambient (60%) and 32oC (52%) conditions (line 280-282). Under ambient condition, the red bar (D1-4) appears to be about 30% at 2mo (Fig. 4b). Please check whether the sentence is correct.
11. The readers might want to know whether bleaching occurred or not during the high temperature treatment.


Small errors
1. Acropora longicyathus instead of longicythus (line 76)
2. Citation of Yamashita et al. (2013)
3. Line 372-375, some words are missing. ‘, which’ can be inserted after ‘Pacific’.
4. Line 477-478. Reference source is not adequate.
5. Change instead of ‘chenge’ (line488)
6. Proceedings. Capitalize the first letter (line 541)

·

Basic reporting

The manuscript titled “shift of symbiont communities in Acropora tennis juveniles under heat stress” by Yorifuji et al” deals with multiple symbiodinium type association in juvenile coral and how this influences their response to high temperature stress.
Authors have carried out the work using the coral A. tenuis, over a period of 2 spawning seasons, effectively repeating the experiment twice.

Specific comments

1. Basic reporting

A. I recommend the authors to check carefully the English again, I find few mistakes in grammar and usage but this can be easily solved.

B. In the introduction, where you introduce Symbiodinium types, please use the scientific names for Symbiodinium that have already been assigned one. For example, type D1a Symbiodinium trenchii

C. Background review and references are sufficient

D. Figures are OK, raw data has been shared

E. I don't see any hypothesis so I cannot comment on that. But the manuscript is self-contained with relevant results

Experimental design

2. Experimental design

The research presented is within the Aims and Scope of the journal

A. I have some difficulty in understanding the experimental design. it would be nice if authors can illustrate their experiment,

B. Why you think 31 ºC is stress temperature? From the value you provided for your ambient seawater temperature range, the highest is already 30.82, and 30.93. How long was this high temperature in the tanks, assuming it to be during summer months? was there any effect on those control juvniles during this time?

C. When you say 250-400 juveniles for each temperature conditions, does that mean that all your 2-3 independent tanks for each temperature had 250-400 nubbins? When i see the raw data, it becomes clear that you pick up random number of juveniles for each tile for each treatment to do the analysis. This is not explain in the main text very clearly

D. In your raw data, for growth and survivorship, you show for each temperature treatment there are 2 sets of 8 tiles from which you picked up certain number of individuals which amounts to more than 30 individuals for each. However, your Symbiodinium data does not have that many number of juveniles analysed for each treatment. Why so?

In the text, line 142, you say 10-20 individuals were collected randomly for the tiles in each tank…..i assume this number is collective sample form 8 tiles, if so, why your raw data shows more than 20 individuals?

E. Line 165 is not clear. Please rewrite it

3. Results

A. You say in 2012, the observation continued up to 1.5 years, however this does not appear in any of the figures. You do have raw data in the supplementary, why?

B. Your results show large difference between 2012 and 2013, why?


C. You used the same parental colonies for both the experiment. Response to temperature (ambient and 30 ºC is different between the years. Do you think this might be due to parental effect? I don’t know hoe about the cross between colonies and which colonies sperm fertilised which colonies egg? Paternal and maternal effect also influences the ability of larvae and juvenile to overcome stress.

Why do you think that the response you observe in juveniles is solely due to Symbiodinium? why not the host itself? irrespective of what Symbiodinium type is present

As you see from your results, there is large variation, especially in 2013, the variation is so big both in prevalence of Symbiodinium types and also D type id always there, so are other types.

D. How come there is no adult Symbiodinium type in juveniles? even at 1.5 years? If so, do you have any idea at what age the juveniles have Symbiodinium same as Adults? - in Okinawa, because as you mention in Discussion (line 375), it is by 1.5 years in the GBR.

D. D1-4 is outdated nomenclature for Symbiodinium. Please update.

Please see the recent publications by LaJeunesse group

Validity of the findings

This will be an important contributions in the field of Symbiodinium shuffling in early life stages of corals and how this might be related to the ability of juveniles to overcome stress. However, I do want to say that, the process of resistance in corals to stress is not solely dependent on the type/types of Symbiodinium it associates with. I feel that the authors opinion is biased towards Symbiodinium effect. Unless it is proven for sure that the Symbiodinium play the sole role in coral stress resistance mechanism, it is not fair to go in this direction.

I am not forcing authors to think in any direction, but may be try to be neutral and put forth both possibilities? there are many papers to support both aspects.

Even from the results of this work, one cannot conclude that Symbidinium shuffling as stress resistance mechanism. Because, the data is from random individuals collected out of hundreds. This might lead to underestimation since the percentage of types in each individual vary.

Also the death of juveniles through time could be due to natural process and finally very few survive and then you see certain Symbiodinium type to be dominant and think that the survival is due to the presence of such type/types, but this might not be the case.

Again from the data presented in this manuscript (line 275), in 2013 50-70% of juveniles at all temperature associated with A3 or C1 in the beginning and then S. trenchii increased under ambient and 32 ºC. This is good example of random changes.


Also when you say that (Line 285) “A greater proportion of coral colonies (this is “Juveniles” and not “colonies” right?) that harboured only S. trenchii survived under 31/32 ºC compared to other treatments”. How do you know this? every time you sampled, you sacrificed the juveniles, there was no way to monitor the type composition in the the juveniles throughout and in the end those survived had S. trenchii, but is it not possible that even those that died also could have been associated with S. trenchii?

So, the reason for increase in the proportion of S. trenchii at later stages of the experiment might have been due to increased mortality of juveniles harbouring different types of Symbiodinium including S. trenchii?

Additional comments

"no comment"

Reviewer 3 ·

Basic reporting

This manuscript addresses the changes in symbiont communities in Acropora tenuis juveniles under heat stress, over a period of four months in one year, and over a period of two months in another year. The study found the prevalence of Symbiodinium type D1-4 and the number of juveniles harboring this type to be higher under heat-stress conditions after 2-4 months.

The authors did a thorough search of the literature to provide a very detailed introduction and the methods were mostly clear and, again, very thorough in providing all the relevant information of how the experiment was set up and how analyses were conducted. There are some spelling and grammatical errors throughout the manuscript, so careful review of the entire document would be necessary.

Experimental design

The experiment was rigorously done, though some continuity between the results in 2012 and 2013 would have been ideal.

Validity of the findings

Given that thorough DGGE analysis had been employed for this study, it would be important to show some a figure with DGGE gel pictures depicting the different patterns associated with the different Symbiodinium types, and which were found in which corals, under which temperature treatments and which time points. Even showing a subset of these data would suffice to give readers an example what was analyzed and how different were the types. The authors can then conveniently refer to this figure under 'Symbiont types in coral juveniles and adults' in the Results. Given that clade F was dropped after this section, it would be informative to see what F looks like in a DGGE gel.

The terms "prevalence" and "occurrence" require careful explanation in both the text and the y-axis labels in the graphs. At first glance of just the figures, it was unclear what was the difference between the two terms. The y-axis labels would be clearer if changed to "Presence of symbiont type (%)" in Fig. 3 and "Proportion of juveniles (%)" in Fig. 4, or something of the like. The clear distinction should be made in the Methods, Results, and Discussion as well.

Fig. 3a shows that type D1-4 is already the most stably prevalent throughout the 4 months at each temperature treatment in 2012 that it would seem difficult to make the case that it is more prevalent at the higher temperatures. The authors can make this case in 2013, however, with the increased prevalence of type D1-4 at 32C in Fig. 3b. But if this conclusion can only be drawn from one set of data but not the other, then there is the question of how repeatable this result is. This would suggest that changes in symbiont communities occur naturally over time, and it is not very conclusive that it is indeed due to temperature alone. The authors also allude to this in lines 331-333, and it is something to carefully consider.

Fig. 4 demonstrates a higher proportion of juveniles harboring type D1-4 at higher temperatures in both years, and some arrows indicating the bars of interest (4-month ambient vs. 4-month 31C in 2012, and 2-month ambient vs. 2-month 32C in 2013) would be helpful and easier for the reader to see.

Fig. 5 could be enhanced with labels of the Symbiodinium clades/types to the right of each shaded cluster. At present, the reader would have to guess/assume that blue clusters to A, red to D, and green to C.

There seems to be some errors in the text where results reported do not match the depiction in the graphs. For example, 63.2% in line 246 is not visible in Fig. 3a, and neither is 60% in line 281 for Fig. 4b. Line 216 should also say Fig. 2a, not 2b.

While the study found the prevalence of Symbiodinium type D1-4 and the number of juveniles harboring this type to be higher under heat-stress conditions after 2-4 months, neither of these aspects seem to help the juveniles survive better or grow larger. It is true that the juveniles that did survive after 2-4 months in 2012 seem to be dominated by type D1-4, but overall survivorship is consistently less than 20% at 2 months, 3 months, and 4 months (Fig. S2a). In other words, survivorship did not improve when juveniles were harboring type D1-4. Therefore, this cannot be considered any type of acclimatization, when these juveniles are clearly not doing so well, and survivorship at 31C is lower than at 30C or ambient temperature. And the 2013 results (Fig. S2b) did not show any type of pattern at all. Hence, it is huge leap to make the conclusion that "type D1-4 increases the heat tolerance of the juveniles" (Discussion, lines 350-351). Additionally, the results at 1.5 years in the 30C treatment should be interpreted with caution, as there is no control treatment to compare that to.

Additional comments

Examining the potential to shift symbiont communities at the coral juvenile stage is important and currently understudied; hence, this attempt by the authors is appreciated and needed in the field. However, the main concern for this manuscript is that the current results do not support the conclusion that clade D Symbiodinium benefit the survivorship of the juveniles under heat stress. The results even slightly suggest the opposite. Therefore, the authors can conclude that (1) type D1-4 is more prevalent in juveniles at the higher temperature in one year and that (2) it occurs in a larger proportion of juveniles at the higher temperature in both years, but that is it. The question of how these two findings could benefit coral juvenile survivorship, growth, and overall health remains inconclusive.

---

## Round 0.2 · accepted · Accept

Dear Dr. Yorifuji and colleagues:

Thank you for taking the time to revise your manuscript, and for addressing all of the concerns of the three reviewers. I am happy to report to you that I believe your work is now suitable for publication in PeerJ. Congratulations! I look forward to seeing this published, and I firmly believe that it will be a valuable contribution to the field. Well done!

-joe